# Low COVID-19 Vaccine Acceptance Is Correlated with Conspiracy Beliefs among University Students in Jordan

**DOI:** 10.3390/ijerph18052407

**Published:** 2021-03-01

**Authors:** Malik Sallam, Deema Dababseh, Huda Eid, Hanan Hasan, Duaa Taim, Kholoud Al-Mahzoum, Ayat Al-Haidar, Alaa Yaseen, Nidaa A. Ababneh, Areej Assaf, Faris G. Bakri, Suzan Matar, Azmi Mahafzah

**Affiliations:** 1Department of Pathology, Microbiology and Forensic Medicine, School of Medicine, The University of Jordan, Amman 11942, Jordan; hananyalu97@gmail.com (H.H.); mahafzaa@gmail.com (A.M.); 2Department of Clinical Laboratories and Forensic Medicine, Jordan University Hospital, Amman 11942, Jordan; 3Department of Translational Medicine, Faculty of Medicine, Lund University, 22184 Malmö, Sweden; 4Department of Dentistry, Jordan University Hospital, Amman 11942, Jordan; deemahameddababseh@gmail.com; 5School of Dentistry, The University of Jordan, Amman 11942, Jordan; hudyeid@gmail.com (H.E.); duaa.taim@yahoo.com (D.T.); ayat_alhaidar@yahoo.com (A.A.-H.); 6School of Medicine, The University of Jordan, Amman 11942, Jordan; kaalhajeri85@icloud.com; 7Department of Clinical Laboratory Sciences, School of Science, The University of Jordan, Amman 11942, Jordan; alaa_mohamad1990@yahoo.com (A.Y.); S.mattar@ju.edu.jo (S.M.); 8Cell Therapy Center (CTC), The University of Jordan, Amman 11942, Jordan; nidaaanwar@gmail.com; 9Department of Biopharmaceutics and Clinical Pharmacy, School of Pharmacy, The University of Jordan, Amman 11942, Jordan; areej_assaf@ju.edu.jo; 10Department of Internal Medicine, School of Medicine, The University of Jordan, Amman 11942, Jordan; fbakri@yahoo.com; 11Infectious Diseases and Vaccine Center, University of Jordan, Amman 11942, Jordan

**Keywords:** vaccine coverage, compulsory vaccination, intention to vaccinate, influenza vaccine, anti-vaxxer, misinformation

## Abstract

Vaccination to prevent coronavirus disease 2019 (COVID-19) emerged as a promising measure to overcome the negative consequences of the pandemic. Since university students could be considered a knowledgeable group, this study aimed to evaluate COVID-19 vaccine acceptance among this group in Jordan. Additionally, we aimed to examine the association between vaccine conspiracy beliefs and vaccine hesitancy. We used an online survey conducted in January 2021 with a chain-referral sampling approach. Conspiracy beliefs were evaluated using the validated Vaccine Conspiracy Belief Scale (VCBS), with higher scores implying embrace of conspiracies. A total of 1106 respondents completed the survey with female predominance (*n* = 802, 72.5%). The intention to get COVID-19 vaccines was low: 34.9% (yes) compared to 39.6% (no) and 25.5% (maybe). Higher rates of COVID-19 vaccine acceptance were seen among males (42.1%) and students at Health Schools (43.5%). A Low rate of influenza vaccine acceptance was seen as well (28.8%), in addition to 18.6% of respondents being anti-vaccination altogether. A significantly higher VCBS score was correlated with reluctance to get the vaccine (*p* < 0.001). Dependence on social media platforms was significantly associated with lower intention to get COVID-19 vaccines (19.8%) compared to dependence on medical doctors, scientists, and scientific journals (47.2%, *p* < 0.001). The results of this study showed the high prevalence of COVID-19 vaccine hesitancy and its association with conspiracy beliefs among university students in Jordan. The implementation of targeted actions to increase the awareness of such a group is highly recommended. This includes educational programs to dismantle vaccine conspiracy beliefs and awareness campaigns to build recognition of the safety and efficacy of COVID-19 vaccines.

## 1. Introduction

The current coronavirus disease 2019 (COVID-19) pandemic can be described as “a once-in-a-century pandemic,” with unparalleled health, social, and economic ramifications [1]. The current number of confirmed COVID-19 cases hit the 100 million mark, with over 2 million fatalities reported to be the consequence of the disease [2].

Hopes to reduce the negative repercussions of COVID-19 were largely dependent on the timely development of efficacious vaccines and their distribution in an equitable manner [3,4]. The expedition of development and approval phases for COVID-19 vaccination trials has culminated in the emergency-use authorization of several safe and effective vaccines—at least in the short term—in a remarkable record time [5,6,7,8,9].

The deployment of COVID-19 vaccines brought a ray of light amid the pandemic crisis darkness. However, the availability of vaccination services is one thing, while the implementation of a successful mass vaccination program is quite another [10]. Challenges that might jeopardize COVID-19 vaccination involve mass manufacturing, global distribution, and cost issues [11]. In addition, ambiguities surrounding some aspects of COVID-19 vaccination include: Uncertainty about long-term protection and the need for frequent reformulation amid recurrent reports of swift severe acute respiratory syndrome coronavirus 2 (SARS-CoV-2) evolution and the emergence of genetic variants [12,13,14,15,16]. Furthermore, COVID-19 vaccine hesitancy appears as a major obstacle in the pandemic control efforts, considering the increasing evidence of its pervasive nature across different regions, countries, and strata of societies [17,18,19,20,21,22,23,24,25,26].

The “delay in acceptance or refusal of vaccines despite availability of vaccination services” defines vaccine hesitancy, which is a complex and context-specific phenomenon that varies across time, place, and vaccines [27,28,29]. It can be triggered by a lack of confidence in the safety and effectiveness of vaccination [30,31]. In addition, inconvenience regarding the vaccine availability, affordability, and accessibility can hinder its acceptance [32]. Moreover, the low perception of disease risk, also termed complacency, can be an inciting factor for vaccine hesitancy [33]. Complacency in the context of COVID-19 vaccine hesitancy is displayed by the lower intent of younger individuals to get vaccinated, which can be correlated with the lower case-fatality rates reported among such a group [18,34].

Notably, the current COVID-19 pandemic can also be described as an “infodemic” [35]. Fear and uncertainty that accompanied COVID-19 crisis fueled the swift spread of misinformation, including the adoption of conspiratorial claims by a considerable proportion of various populations [23,36,37]. Such conspiracy theories extended to involve vaccination aspects like the bizarre hoax that COVID-19 vaccination is a scheme to implant microchips or quantum-dot spy software for monitoring purposes [23,38]. Other ludicrous claims included suggestions that SARS-CoV-2 is man-made and the allegations of a link between messenger RNA vaccines and infertility [39,40].

Since we are living in the era of social media, viral misinformation might be one of the biggest pandemic risks [41,42]. This was illustrated previously by the larger embrace of conspiratorial claims among people relying on social media platforms to get knowledge on various aspects of the pandemic [23,37].

Conspiracy beliefs were previously linked to abstinence from health-related behaviors (e.g., vaccination, contraceptive behaviors) [43]. In a similar vein, the association between COVID-19 vaccine conspiracy beliefs and the low intent to get COVID-19 vaccines was demonstrated in our previous study among the general public in Arab countries [23].

In Jordan, the total number of reported COVID-19 cases was approximately 321,000, with 4239 mortalities linked to the disease as of January 26, 2021 [2]. Since university students can be viewed as an insightful group of younger individuals, assessing their attitude towards COVID-19 vaccination requires special attention. Thus, the aims of the current study were as follows: To evaluate the overall intent to get COVID-19 and influenza vaccines among university students in Jordan; to assess the possible factors associated with higher COVID-19 vaccine hesitancy among the study group; to examine the potential correlation between vaccine conspiracy beliefs and COVID-19 vaccine hesitancy; and to assess the association between the key sources of knowledge about COVID-19 vaccines and its acceptance.

## 2. Materials and Methods

### 2.1. Study Design

This cross-sectional study was conducted using an online-based survey that took place between 19 January 2020 (21:00) and 23 January 2020 (21:00). Recruitment of the participants was based on a chain-referral sampling approach, starting with contacts of the authors and sharing the survey on social media platforms (Facebook, Twitter, and Instagram), besides the free messaging service; WhatsApp. The target population eligible for inclusion in this study was university students currently studying in Jordan and aged 18 or over.

The current data points to about 300,000 university students in Jordan, that are affiliated with 32 Universities (10 of which are public, and 22 are private) [44]. Based on the aforementioned figure and considering a margin of error equaling 3.0% (with 95% confidence interval), the minimum calculated sample size was 1064 respondents [45].

### 2.2. Overview of Survey Items

This study was based on a survey comprising four sections with 23 items (Appendix A). The 1st section assessed students’ demographics and previous experience with COVID-19 and included questions on the following: Age, sex, educational level, history of any chronic disease (such as diabetes, allergy, hypertension, or heart disease), and previous COVID-19 diagnosis in the respondent or a family member.

The 2nd section comprised items that assessed: The belief in conspiracy about COVID-19 origin, the belief that SARS-CoV-2 was manufactured to force the public to get vaccinated, willingness to get COVID-19 vaccines, willingness to get influenza vaccine, opposition to vaccination in general, the belief that COVID-19 vaccine is a way to implant microchips into people as a control scheme, and the belief that COVID-19 vaccines will lead to infertility.

The 3rd section assessed the single main source of knowledge about COVID-19 vaccines (allowing selection of a single main source out of 3 possible options: 1. Television and news releases; 2. Social media platforms/YouTube (Facebook, Twitter, WhatsApp among others); 3. Medical doctors, scientists, or scientific journals).

Finally, the 4th section was based on the brief previously validated Vaccine Conspiracy Beliefs Scale (VCBS), with minor modifications to accommodate questions on COVID-19 vaccines [23,46]. Response to all items before submission was mandatory to eliminate the effects of item non-response.

### 2.3. Measures

#### 2.3.1. Intention to Get COVID-19 Vaccines

The major outcome measure in this study was the willingness to get COVID-19 vaccines, with responses dichotomized as: Yes vs. no/maybe.

#### 2.3.2. Assessment of COVID-19 Vaccine Conspiracy Beliefs

The correlation between conspiracy beliefs regarding COVID-19′s origin and vaccine conspiratorial claims in relation to COVID-19 vaccine acceptance was evaluated using multinomial logistic regression.

#### 2.3.3. Covariates in Multinomial Regression Analysis

The covariates used included: Age (≤21 years vs. >21 years); sex; nationality (Jordanian vs. non-Jordanian); University (public vs. private); School/Faculty (Health/Scientific vs. Humanities); educational level (undergraduate vs. postgraduate); history of chronic disease; previous self/family experience of COVID-19.

#### 2.3.4. Vaccine Conspiracy Beliefs Scale (VCBS)

Using a 7-point scale, the respondents were asked to indicate how much they agreed or disagreed with 7 statements to evaluate their views on vaccine conspiracy. A response with “strongly disagree” was given the minimum score of 1, and the maximum score of 7 was given to “strongly agree” response resulting in a direct relationship between VCBS and the embrace of COVID-19 vaccine conspiracies. Internal consistency of the VCBS was ensured by Cronbach’s alpha value of 0.948

#### 2.3.5. Intention to Get COVID-19 Vaccines in Relation to VCBS

This correlation was evaluated using univariate analysis with the intention for COVID-19 vaccination as the dependent variable, VCBS as the fixed factor, and the following as covariates: Sex; nationality; University; School/Faculty.

### 2.4. Ethical Considerations

The study was approved by the Department of Pathology, Microbiology and Forensic Medicine (meeting 03/2020/2021) and by the Scientific Research Committee at the School of Medicine/University of Jordan (reference number: 321/2021/67). Informed consent was ensured by the presence of an introductory section of the survey used in this study, with a mandatory question asking for agreement from the respondent to participate in the survey study. All collected data were treated with confidentiality.

### 2.5. Statistical Analysis

To characterize the study variables, we used measures of central tendency (mean) and dispersion (standard deviation (SD)). Associations between categorical variables were assessed using the chi-squared test (χ^2^). Two-tailed Mann–Whitney *U* test was used to assess the association between scale variables (age, VCBS) and binary categorical variables. Univariate and multinomial regression analyses were used as appropriate. The statistical significance was considered for *p* <0.050, and all analyses were conducted using IBM SPSS Statistics for Windows, Version 22.0. Armonk, NY: IBM Corp.

## 3. Results

### 3.1. Characteristics of Survey Respondents

The total number of respondents included in final analysis was 1106 students. Characteristics of the respondents divided by academic discipline (Health, Scientific, Humanities) are presented in (Table 1). The participant students belonged to a total of 24 universities; with a majority being affiliated to the University of Jordan (*n* = 605, 54.7%), followed by Mutah University (*n* = 120, 10.8%), Hashemite University (*n* = 70, 6.3%), Jordan University of Science and Technology, and Al-Balqa’ Applied University (*n* = 69, 6.2% for each). For the Schools/Faculties with ≥25 participants, the distribution was: Medicine (*n* = 342, 30.9%), Pharmacy (*n* = 191, 17.3%), Engineering (*n* = 92, 8.3%), Science (*n* = 80, 7.2%), Dentistry (*n* = 69, 6.2%), Agriculture (*n* = 59, 5.3%), Arts (*n* = 41, 3.7%), Law (*n* = 33, 3.0%), Business (*n* = 31, 2.8%), and Information Technology (*n* = 25, 2.3%).

### 3.2. Low Intent to Get COVID-19 Vaccines among the Respondent Students

The overall intent to get COVID-19 vaccination among the respondents was as follows: Yes (*n* = 386, 34.9%), no (*n* = 438, 39.6%), and maybe (*n* = 282, 25.5%). Variables that were associated with a higher intent to get COVID-19 vaccines are summarized in Table 2. The variables associated with a higher intent for getting COVID-19 vaccines included: Male sex, non-Jordanian nationality, and affiliation to a Health School or a public University.

### 3.3. Low Acceptance of Influenza Vaccines among the Respondent Students

The overall acceptance of influenza vaccination among the respondents was as follows: Yes (*n* = 318, 28.8%), no (*n* = 578, 52.3%), and maybe (*n* = 210, 19.0%). Variables that were associated with a higher acceptance of influenza vaccination are summarized in Table 3 and included: Younger age, non-Jordanian nationality, affiliation to a Health School, and a previous history of chronic disease. The majority of students who accepted influenza vaccination showed an intent to get COVID-19 vaccines (192/318, 60.4%), and the majority of respondents with influenza vaccine hesitancy (no/maybe) displayed a hesitancy towards COVID-19 vaccination as well (594/788, 75.4%; *p* <0.001, χ^2^ test).

### 3.4. The Belief in Conspiratorial Claims Was Associated with Lower COVID-19 Vaccine Acceptance

Among the study respondents, the overall belief that COVID-19 is a man-made disease was 29.7% (with 25.4% having no opinion). Additionally, 11.0% of the respondents stated that COVID-19 was man-made to enforce vaccination (with 36.0% who responded with maybe). Moreover, 73.6% of the respondents rejected the claim that COVID-19 vaccination will be used to implant microchips into humans to control them, and 54.6% rejected the claim that COVID-19 vaccination can lead to infertility.

The association between COVID-19 vaccine hesitancy (intent to get COVID-19 vaccines with no/maybe responses) and conspiratorial claims was evaluated using multinomial logistic regression. These five conspiratorial claims included: COVID-19 is a man-made disease; COVID-19 was manufactured to enforce vaccination; COVID-19 vaccination intends to implant microchips into people to control them; COVID-19 vaccination will lead to infertility; and general opposition to vaccination (anti-vaccination). The covariates used included: Age (≤21 years vs. >21 years); sex; nationality (Jordanian vs. non-Jordanian); University (public vs. private); School/Faculty (Health/Scientific vs. Humanities); educational level (undergraduate vs. postgraduate); history of chronic disease; and previous self/family experience of COVID-19. Declining to reject each of the five conspiratorial claims was associated with a statistically significant higher likelihood of COVID-19 vaccine hesitancy (Figure 1).

### 3.5. Vaccine Conspiracy Beliefs Were Associated with a Significantly Higher Level of COVID-19 Vaccine Hesitancy

The previously validated VCBS was used to evaluate the association between vaccine conspiracy beliefs and COVID-19 vaccine hesitancy. A statistically significant difference was found between the VCBS among the respondents with intent to get COVID-19 vaccines (mean = 15.9, SD = 7.7), compared to the hesitant respondents (mean = 27.0, SD = 9.8, Figure 2). This difference was supported by an actual *p*-value of 2.3 × 10^−65^. Additionally, univariate analysis with the intention for COVID-19 vaccination as the dependent variable, VCBS as the fixed factor, and the following as covariates: Sex; nationality; University; and School/Faculty, showed that a higher VCBS was associated with COVID-19 vaccine hesitancy (*p* < 0.001).

### 3.6. Dependence on Social Media Platforms Was Associated with COVID-19 Vaccine Hesitancy

The overall intention to get COVID-19 vaccines was the highest among students who reported dependence on medical doctors, scientists, and scientific journals for knowledge regarding the vaccine (47.2%). This was followed by dependence on TV programs and news releases (26.0%), while the lowest rate of intention to get the vaccine was among those who depended on social media platforms (19.8%; *p* < 0.001, χ^2^). The dependence on social media platforms was also associated with a higher prevalence of influenza vaccine hesitancy and with a general opposition to vaccination, compared to dependence on medical doctors, scientists, and scientific journals (*p* = 0.040 and *p* < 0.001 for the comparisons, respectively, χ^2^, Figure 3).

## 4. Discussion

Results of the current study clearly showed one harmful effect of embracing COVID-19 conspiracy beliefs; namely vaccine hesitancy among university students at a country level. The conspiracy rhetoric might seem harmless. However, more evidence is accumulating that points to its potential threat, including poor health behavior, besides the detrimental social and psychologic effects [36,37,40,47,48].

University students constitute a presumably knowledgeable and aware group of society, with a more open attitude. Hence, students could have a leading role in public service. During the current COVID-19 era, this role is recommended particularly for university students in Health Schools/Faculties, through the promotion of clear scientific-based helpful messages (e.g., the role of vaccination as a cornerstone in public health) [49,50]. In addition, previous evidence showed that university students can comprise a core group that would be helpful in addressing vaccine hesitancy through promoting a positive attitude towards vaccination [51].

Other potential roles that students can play during the current pandemic entail informing peers/relatives about the importance of preventive measures (e.g., use of masks and physical distancing) [49]. In addition, university students can help to identify and rectify falsified messages about various aspects of COVID-19, including those related to vaccination. This role is of particular importance through social media platforms, considering the younger demographics of social media users [52]. Thus, the evaluation of the students’ baseline level of knowledge and attitude towards vaccination is necessary to identify potential defects that may negatively impact their helpful role.

In light of the objectives of the current study, the main results can be summarized as follows:

First, university students in Jordan showed an overall low intent to get COVID-19 vaccines (34.9%). This alarming result is in agreement with our recent survey that investigated COVID-19 vaccine hesitancy among the general public in Arab countries, where the vaccine acceptance rate was merely 28.4% in Jordan [23]. Further analysis of COVID-19 vaccine hesitancy in this study showed a relatively higher vaccine acceptance rate among students in Health Schools (43.5%), compared to their peers in Scientific or Humanities Schools (23.6%). This can be related to their higher knowledge about the disease [40]. Additionally, students in Health Schools might have a better ability to fathom the results of clinical trials on COVID-19 vaccines; thus, having a higher trust and acceptance of such novel vaccines. Complacency can be a factor behind such a low intent to get vaccinated, given the findings of lower morbidity and mortality of COVID-19 among younger individuals. Nevertheless, the control of COVID-19 spread in communities, and lessening its devastating effects, cannot be achieved without the cooperation of such a group. Low perception of disease risk can be an important determinant in the declining intention to get the vaccines, as shown recently by Caserotti et al. [53]. Unraveling such an attitude would be highly valuable to tailor communication regarding COVID-19 vaccination [54]. This would be particularly relevant in countries with a prominent youth bulge (e.g., Jordan, where 20% of its population aged 20–29 in 2019) [55].

To put the previous result in a broader perspective, we found a few studies that investigated COVID-19 vaccine hesitancy among university students [17]. A higher rate of COVID-19 vaccine acceptance (57.3%) was reported by Gretch and Gauci in Malta among Health Sciences, Dentistry, and Medicine students [22]. This result can be used to advocate the role of university students in Health Schools (e.g., medical, dental, pharmacy, and nursing students) in the dissemination of correct messages regarding vaccination [56]. These helpful messages can be disseminated both among their colleagues and the general public as well. In Italy, Barello et al. reported a much higher rate of intention to get COVID-19 vaccines (86.1%) and even a higher rate of vaccine acceptance among students at Health Schools [57]. Additionally, a recent report from Michigan showed that about one-quarter of the medical students displayed COVID-19 vaccine hesitancy [56]. Thus, our results might indicate a negative attitude towards these novel vaccines among university students in Jordan.

Second, male sex and non-Jordanian nationality were found to be associated with a higher intent to get COVID-19 vaccines. In our previous studies, males were found to have perceived COVID-19 as a more dangerous disease compared to females, besides their lower tendency to believe in conspiracies and their lesser reliance on social media platforms to get knowledge about the disease [23,40]. For non-Jordanian students, being abroad and away from their families besides the higher anxiety levels might cause them to feel more inclined to accept vaccination as a protective measure [40].

Third, a major result of this study was the independent correlation between the belief in conspiracy and COVID-19 vaccine hesitancy among university students. To elaborate on this result, the role of misinformation cannot be overlooked [58]. Dismantling conspiracy beliefs requires a special focus on delivering clear, timely, and evidence-based messages through legitimate channels. The suggested measures to mitigate misinformation involve the collaboration of the scientific community/experts and media sources [59]. Another role relies on social media companies through vigilant fact-checking and flagging of content that spreads misinformation [60,61]. A recurring pattern among various previously studied populations was replicated in this study, which is the association of dependence on social media platforms with the embrace of conspiracy beliefs. Thus, university students can play a major role in the fight against misinformation, provided they are given scientific evidence backed by rational thinking to tackle such a phenomenon.

Fourth, a worrying result was that vaccine hesitancy involved the influenza vaccine and vaccination in general. Considering the potential impact of such a view among the future parents and guardians, childhood vaccination might be at risk with the potential resurgence of some infectious diseases instead of the aspiration to be eradicated.

Limitations of this study involve potential sampling bias that could be related to coverage of the students who use the internet and social media platforms regularly, besides the limited willingness to participate in using an online-based survey, which prevented the inclusion of a larger sample size. Despite the advantages of the chain-referral approach in the sampling of hidden populations, an obvious caveat of this approach is that bias introduced by the first accessed participants would be inevitable. Thus, the generalizability of the results cannot be ensured. The higher proportion of responses from Health Schools/Faculties could have resulted in a bias towards higher rates of vaccine acceptance. An additional limitation was the absence of questionnaire items to assess risk perception among the participants thoroughly. The inclusion of such items could have added an additional perspective to this study, considering the potential role of risk perception in vaccine acceptance [62]. Thus, future studies tackling similar aims should focus on such aspects in survey design.

Moreover, it should be stressed that vaccine hesitancy is a multifactorial phenomenon [63]. Thus, holding conspiracy beliefs was correlated with COVID-19 vaccine hesitancy in this study but cannot solely explain the non-intent to get the vaccines among the participants who did not embrace conspiracy beliefs. Such a negative attitude might be related to complacency or lack of confidence in the safety and effectiveness of the novel COVID-19 vaccines [64]. An additional important factor that should not be overlooked in the context of COVID-19 vaccine hesitancy is the role of mistrust in policymakers, healthcare professionals, and vaccine providers [65]. These factors should be considered in further studies addressing vaccine hesitancy in the country.

## 5. Conclusions

In 2018, a centennial perspective reflection on the Spanish flu pandemic was made by Heidi J. Larson, the founding director of the Vaccine Confidence Project [41]. In her article entitled “The biggest pandemic risk? Viral misinformation”, she warned against vaccine hesitancy harms if any future pandemic takes place, and in less than two years, it appeared that viral misinformation materialized faster than anyone could have imagined.

The harmful and dangerous embrace of conspiracy beliefs that surrounded the current pandemic extended to affect the intent to get COVID-19 vaccination even among university students, supposedly a well-educated and insightful part of society. Only one-third of university students in Jordan showed a clear intent to get COVID-19 vaccines, which is an alarming rate that could hamper the preventive control efforts in the country.

Another result that should be considered carefully was the considerable proportion of university students who reported being anti-vaccination altogether. This warrants interventional measures to build recognition of the importance of vaccination from a public health point of view. Finally, social media companies are recommended to take a powerful role in enforcing fact-checking in a better way to counteract the dissemination of misinformation.

## Figures and Tables

**Figure 1 ijerph-18-02407-f001:**
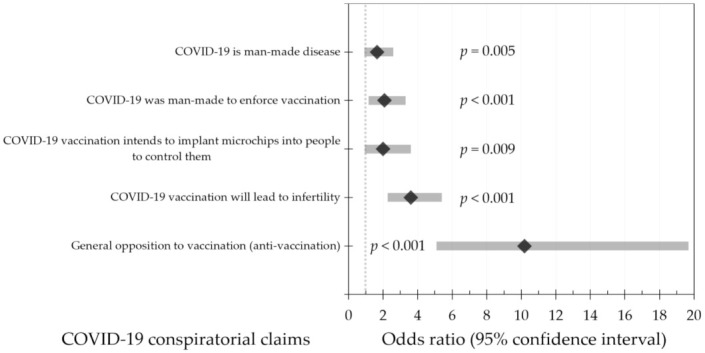
Multinomial regression analysis of the five conspiratorial claims and their association with COVID-19 vaccine hesitancy (intent to get COVID-19 vaccination with no/maybe responses). The mean odds ratio is represented by the diamond shape, while the 95% confidence interval is displayed as the grey bar.

**Figure 2 ijerph-18-02407-f002:**
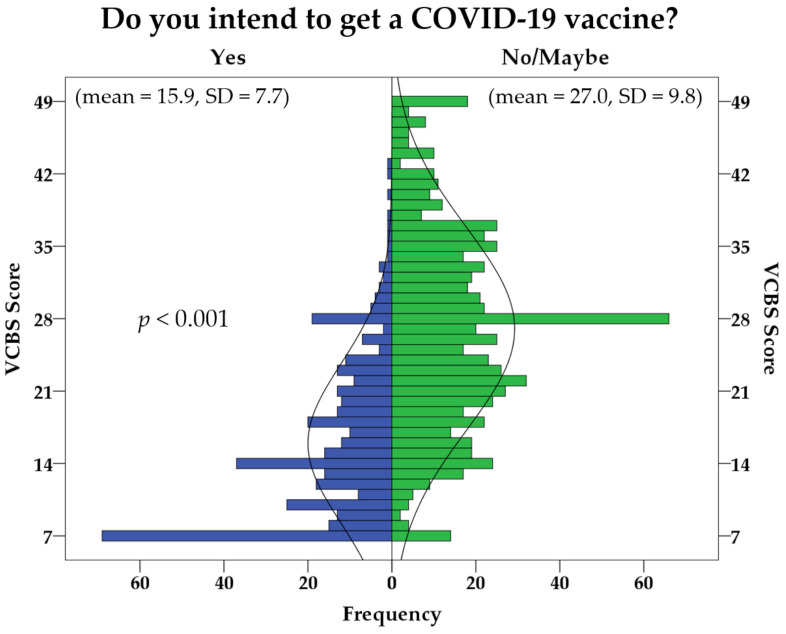
The vaccine conspiracy belief score (VCBS) and its association with higher COVID-19 vaccine hesitancy (intent to get COVID-19 vaccination with no/maybe responses). SD: Standard deviation.

**Figure 3 ijerph-18-02407-f003:**
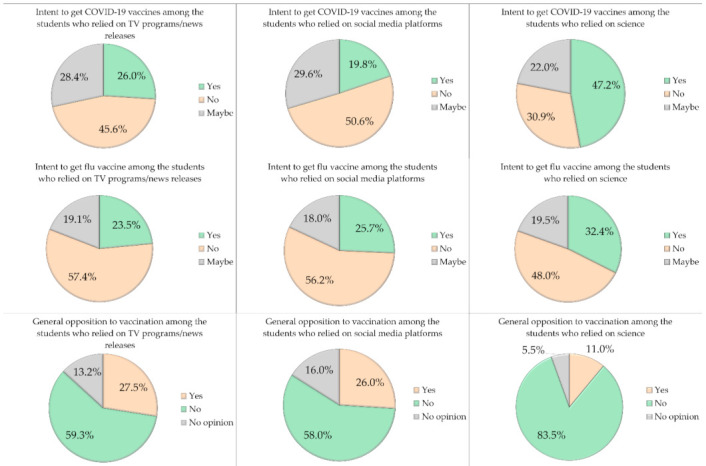
The intent to get COVID-19 vaccines, influenza vaccine and general opposition to vaccination, stratified by the main source of knowledge regarding COVID-19 vaccination.

**Table 1 ijerph-18-02407-t001:** Characteristics of survey respondents divided by their academic discipline.

Characteristic	Academic Discipline	School/Faculty	School/Faculty	School/Faculty
	Feature	Health n ^2^ (%)	Scientific *n* (%)	Humanities *n* (%)
Mean age (SD ^1^)		20.8 (2.7)	22.5 (4.4)	23.9 (6.6)
Age categories	≤21 years	473 (75.3)	136 (49.6)	97 (47.5)
	>21 years	155 (24.7)	138 (50.4)	107 (52.5)
Nationality	Jordanian	471 (75.0)	253 (92.3)	188 (92.2)
	Non-Jordanian	157 (25.0)	21 (7.7)	16 (7.8)
University	Public	582 (92.7)	249 (90.9)	149 (73.0)
	Private	46 (7.3)	25 (9.1)	55 (27.0)
Sex	Male	168 (26.8)	82 (29.9)	54 (26.5)
	Female	460 (73.2)	192 (70.1)	150 (73.5)
Educational level	Undergraduate	596 (94.9)	225 (82.1)	149 (73.0)
	Postgraduate	32 (5.1)	49 (17.9)	55 (27.0)
History of chronic disease	Yes	57 (9.1)	28 (10.2)	20 (9.8)
	No	571 (90.9)	246 (89.8)	184 (90.2)
Experience of COVID-19 in self or family	Yes	252 (40.1)	103 (37.6)	69 (33.8)
	No	376 (59.9)	171 (62.4)	135 (66.2)

^1^ SD: Standard deviation. ^2^
*n*: Number.

**Table 2 ijerph-18-02407-t002:** Analysis of respondent variables for possible association with intent to get COVID-19 vaccines.

Variable	Feature	Intent for COVID-19 Vaccination		*p*-Value ^3^
		Yes	No/Maybe	
		*n*^2^ (%)	*n* (%)	
Mean age (SD ^1^)		21.4 (3.6)	21.9 (4.6)	0.215
Age categories	≤21 years	256 (36.3)	450 (63.7)	0.207
	>21 years	130 (32.5)	270 (67.5)	
Nationality	Jordanian	289 (31.7)	623 (68.3)	<0.001 **
	Non-Jordanian	97 (50.0)	97 (50.0)	
University	Public	359 (36.6)	621 (63.4)	0.001 **
	Private	27 (21.4)	99 (78.6)	
School/Faculty	Health	273 (43.5)	355 (56.5)	<0.001 **
	Scientific	64 (23.4)	210 (76.6)	
	Humanities	49 (24.0)	155 (76.0)	
Sex	Male	128 (42.1)	176 (57.9)	0.002 **
	Female	258 (32.2)	544 (67.8)	
Educational level	Undergraduate	339 (34.9)	631 (65.1)	0.929
	Postgraduate	47 (34.6)	89 (65.4)	
History of chronic disease	Yes	35 (33.3)	70 (66.7)	0.723
	No	351 (35.1)	650 (64.9)	
Experience of COVID-19 in self or family	Yes	150 (35.4)	274 (64.6)	0.793
	No	236 (34.6)	446 (65.4)	

^1^ SD: Standard deviation. ^2^
*n*: Number. ^3^
*p*-value: For age comparisons, we used Mann–Whitney *U* tests, and for categorical variables, we used chi-squared tests. The two asterisks were used to highlight *p*-values ≤ 0.010.

**Table 3 ijerph-18-02407-t003:** Analysis of respondent variables for possible association with acceptance of influenza vaccination.

Variable	Feature	Acceptance of Influenza Vaccination		*p*-Value ^3^
		Yes	No/Maybe	
		*n*^2^ (%)	*n* (%)	
Mean age (SD ^1^)		21.2 (3.5)	21.9 (4.6)	0.033 *
Age categories	≤21 years	219 (31.0)	487 (69.0)	0.027 *
	>21 years	99 (24.8)	301 (75.3)	
Nationality	Jordanian	249 (27.3)	663 (72.7)	0.021 *
	Non-Jordanian	69 (35.6)	125 (64.4)	
University	Public	287 (29.3)	693 (70.7)	0.274
	Private	31 (24.6)	95 (75.4)	
School/Faculty	Health	212 (33.8)	416 (66.2)	<0.001 **
	Scientific	56 (20.4)	218 (79.6)	
	Humanities	50 (24.5)	154 (75.5)	
Sex	Male	95 (31.3)	209 (68.8)	0.259
	Female	223 (27.8)	579 (72.2)	
Educational level	Undergraduate	285 (29.4)	685 (70.6)	0.217
	Postgraduate	33 (24.3)	103 (75.7)	
History of chronic disease	Yes	40 (38.1)	65 (61.9)	0.026 *
	No	278 (27.8)	723 (72.2)	
Experience of COVID-19 in self or family	Yes	136 (32.1)	288 (67.9)	0.054
	No	182 (26.7)	500 (73.3)	

^1^ SD: Standard deviation. ^2^
*n*: Number. ^3^
*p*-value: For age comparisons, we used Mann–Whitney *U* tests, and for categorical variables, we used chi-squared tests. The single asterisk was used to highlight *p*-values *p* ≤ 0.050, while the two asterisks were used to highlight *p*-values ≤ 0.010.

## Data Availability

The data presented in this study are available on request from the corresponding author (M.S.).

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
