# Peer review of "Low COVID-19 Vaccine Acceptance Is Correlated with Conspiracy Beliefs among University Students in Jordan"

_ijerph, 2021, doi:10.3390/ijerph18052407_

Round 1
Reviewer 1 Report
The paper deals with intentions to vaccinate against covid-19 in Jordan (and, more marginally, intentions to vaccinate against the flu). It focuses on the role of conspiracy beliefs. Its main interest rests in documenting vaccine hesitancy in a country seldom commented on in the international litterature.
That said, I believe a number of limitations must be adressed before publication.
- From a methodological standpoint, the main limitations of this paper is that the questionnaire did not integrate items on the perception of the threat posed by the disease. This is crucial given that so many doubt the severity of this disease but also because the actual threat posed by this virus varies depending on many factors, including age. It wouldn’t be surprising that young people do not perceive the virus as particularly harmful to them (and studies have already documented this for other countries). Risk perception is one of the main drivers of non-vaccination. The fact that this isn’t integrated in the explanation of intentions to vaccinate against covid is therefore problematic, especially in a context where it is very uncertain whether COVID-19 vaccines prevent contamination of others (the survey was conducted in january). I don’t think this warrants rejection of the paper. But I do think this limitation should be adressed explicitly in the paper and that more space should be given to this aspect in the interpretation of the results.
- The paper tends to focus on conspiracy beliefs to explain low intentions to vaccinate. It shows that when people adhere to certain conspiracy beliefs, they tend to refuse COVID-19 vaccination. The scale they use has been tested in previous studies and does (i believe) focus on beliefs that can be labelled as « conspiracy theories ». But It is unclear how much of vaccine refusal can be explained by these beliefs. We see clearly that when people hold these beliefs they refuse vaccination, but how much of the 39,6% refusal and 25,5% maybe can be explained by these beliefs ? In addition to statistical tests, a simple first step would be to provide the proportion of refusers who agree with these conspiracy beliefs (with the degree of approval) but also the proportion of those who answered « maybe » or « yes ». I think by doing so, we would also find that many people who hold these beliefs (or « somewhat agree » with them) would get vaccinated. This leads me to my 3rd point.
- The authors tend to equate refusal to be vaccinated (and « maybes » are often lumped in with the nos) with belief in conspiracies. This can be seen in the introduction and discussion sections where the only elements brought forward to understand why people might be reluctant to be vaccinated are misinformation, conspiracy theories, etc. Any form of doubt seems to be equated with the irrationnal rejection of science and public authorities. This may not be what the authors have wished to convey, but this is the message that came accross to me when I read it. On the contrary, the litterature on vaccine hesitancy has underlined the many reasonable reasons people can have to distrust authorities (ranging from discriminations, past scandals, politicization of vaccines…). I suggest the authors read Heidi Larson’s review of the literature on the role of trust (in Human Vaccines and Immunotherapeutics). Not all form of distrust of authorities constitutes conspiracy beliefs. The term trust does not appear even once in this paper. This should be especially clear in the context of the COVID pandemic : some governments’ actions have given good reasons to be cautious : Trump politicized the issue, the UK government decided to space the second dose against scientific advice, the contracts signed between the EU and pharmaceutical companies have not been made public… This focus on conspiracy beliefs can come from the fact that the questionnaire did not offer any items where respondents could express forms of distrust of various actors involved in vaccine production and rollout in Jordan (apart for the conspiracy beliefs discussed previously). This should be clearly adressed.
- More information should be provided on the history/context of vaccine hesitancy in Jordan before COVID-19 but also on the debates over COVID-19 vaccination. Have their been major public debates on the safety of these vaccines in the Jordan press ? Are there important antivaccine movements in Jordan and have they made themselves heard since the beginning of the pandemic ? This is important to interpret the results. In the discussion the authors write this : « First, University students in Jordan showed an overall low intent to get COVID-19 287 vaccines (34.9%). This alarming result is in agreement with our recent survey that inves-288 tigated COVID-19 vaccine hesitancy among the general public in Arab countries, where 289 the vaccine acceptance rate was merely 28.4% in Jordan [23]. » This should be developped so we can get a grasp of the specificities of attitudes towards vaccines in Jordan. Indeed, vaccine hesitancy is very context specific, and providing more context on Jordan would be -in itself – a welcome addition to the littérature.
Here are some more minor comments :
- The definition of vaccine hesitancy provided in the introduction (« The refusal to accept vaccination in spite of effective and safe vaccine availability 71 defines vaccine hesitancy ») does not correspond to the standard in the littérature. Hesitant people often do get vaccinated and hesitancy also arises in contexts where there is uncertainty regarding the effectiveness and safety of a given vaccine (as is the case regarding COVID-19 vaccination). I suggest the authors turn to the standard definitions provided by larson, dubé or Peretti-Watel.
- I would strengthen the justification for focusing on students. In the paper, they are presented as « insightful », etc. I don’t think this really underlines the importance of studying this sub-group. In the discussion, the authors mention their potential influence on fellow young people : this could be expanded (especially since this is more likely to be the case for medical students than for students of the humanities, but I’m not familiar with social status in Jordan). Other justifications could be that young people tend to be more reluctant to vaccinate against covid and that it’s important to understand why because eventually we will need to get them vaccinated to avoid the spread of the virus.
- I think a little bit more should be said about the potential biases due to the chain-referral sampling
Author Response
Reviewer #1 comments
The paper deals with intentions to vaccinate against covid-19 in Jordan (and, more marginally, intentions to vaccinate against the flu). It focuses on the role of conspiracy beliefs. Its main interest rests in documenting vaccine hesitancy in a country seldom commented on in the international litterature.
That said, I believe a number of limitations must be adressed before publication.
- From a methodological standpoint, the main limitations of this paper is that the questionnaire did not integrate items on the perception of the threat posed by the disease. This is crucial given that so many doubt the severity of this disease but also because the actual threat posed by this virus varies depending on many factors, including age. It wouldn’t be surprising that young people do not perceive the virus as particularly harmful to them (and studies have already documented this for other countries). Risk perception is one of the main drivers of non-vaccination. The fact that this isn’t integrated in the explanation of intentions to vaccinate against covid is therefore problematic, especially in a context where it is very uncertain whether COVID-19 vaccines prevent contamination of others (the survey was conducted in january). I don’t think this warrants rejection of the paper. But I do think this limitation should be adressed explicitly in the paper and that more space should be given to this aspect in the interpretation of the results.
Response: We would like to thank the reviewer for this important comment. We have briefly touched upon the role of complacency and the low perception of COVID-19 risk among younger individuals (including university students) in relation to the intention to get COVID-19 vaccines. (Please refer to the Introduction section, Page 2, Lines 76-80; Discussion section, Page 10, Lines 308-311). However, based on the valuable suggestion made by the reviewer, the following changes were made to elaborate more on this important subject:
Please refer to the Discussion section, Page: 10, lines: 311-316:
“Low perception of disease risk can be an important determinant in the declining intention to get the vaccines, as shown recently by Caserotti et al [51]. Unravelling such an attitude would be highly valuable to tailor communication regarding COVID-19 vaccination [52]. This would be particularly relevant in countries with a prominent youth bulge (e.g. Jordan, where 20% of its population aged 20-29 in 2019) [53].”
In the limitations sub-section, the following paragraph was added as well (Page 11, Lines 360-365):
“An additional limitation was the absence of questionnaire items to thoroughly assess risk perception among the participants. The inclusion of such items could have added an additional perspective to this study, considering the potential role of risk perception in vaccine acceptance [62]. Thus, future studies tackling similar aims should focus on such aspect in survey design.”
- The paper tends to focus on conspiracy beliefs to explain low intentions to vaccinate. It shows that when people adhere to certain conspiracy beliefs, they tend to refuse COVID-19 vaccination. The scale they use has been tested in previous studies and does (i believe) focus on beliefs that can be labelled as « conspiracy theories ». But It is unclear how much of vaccine refusal can be explained by these beliefs. We see clearly that when people hold these beliefs they refuse vaccination, but how much of the 39,6% refusal and 25,5% maybe can be explained by these beliefs ? In addition to statistical tests, a simple first step would be to provide the proportion of refusers who agree with these conspiracy beliefs (with the degree of approval) but also the proportion of those who answered « maybe » or « yes ». I think by doing so, we would also find that many people who hold these beliefs (or « somewhat agree » with them) would get vaccinated. This leads me to my 3rd point.
Response: We would like to thank the reviewer for raising this important issue. One of the major aims of this study was to examine the potential correlation between vaccine conspiracy beliefs and COVID-19 vaccine hesitancy. Clearly, the vaccine hesitancy phenomenon is multifaceted with several factors that could have a role in the negative attitude towards COVID-19 vaccination (which was mentioned in the Introduction section: “The “delay in acceptance or refusal of vaccines despite availability of vaccination services” defines vaccine hesitancy, which is a complex and context-specific phenomenon that varies across time, place and vaccines [27-29]. It can be triggered by lack of confidence in the safety and effectiveness of vaccination [30,31]. In addition, inconvenience regarding the vaccine availability, affordability and accessibility can hinder its acceptance [32]. Moreover, the low perception of disease risk, also termed complacency, can be an inciting factor for vaccine hesitancy [33]. Complacency in the context of COVID-19 vaccine hesitancy, is displayed by the lower intent of younger individuals to get vaccinated, which can be correlated with the lower case-fatality rates reported among such a group [18,34].”
Thus, we aimed to examine one aspect of the potential motives driving vaccine hesitancy among University students in Jordan, which was related to the role of misinformation (particularly conspiracy theories surrounding the origin of the virus and vaccinations aspects).
Considering the importance of the reviewer comment, we added the following statement to the Discussion section (Page 11, Lines 366-371): “Moreover, it should be stressed that vaccine hesitancy is a multifactorial phenome-non [63]. Thus, holding conspiracy beliefs was correlated with COVID-19 vaccine hesitancy in this study, but cannot solely explain non-intent to get the vaccines among the participants who did not embrace conspiracy beliefs. Such a negative attitude might be related to complacency or lack of confidence in safety and effectiveness of the novel COVID-19 vaccines [64].”
Also, we aimed to assess COVID-19 vaccine acceptance vs. reluctance/rejection/hesitancy grouped together. Based on the previous explanation, we prefer to keep the stratification of the item on the intent to get the COVID-19 vaccines as (yes vs. no/maybe) in the current state since we wanted to assess vaccine acceptance vs. hesitancy. Such an approach was used previously even on scales as you can find in the following references (La Vecchia et al, 2020, Italy; Freeman et al, 2020, UK; Reiter et al, 2020, US; Lin et al, 2020, China).
- The authors tend to equate refusal to be vaccinated (and « maybes » are often lumped in with the nos) with belief in conspiracies. This can be seen in the introduction and discussion sections where the only elements brought forward to understand why people might be reluctant to be vaccinated are misinformation, conspiracy theories, etc. Any form of doubt seems to be equated with the irrationnal rejection of science and public authorities. This may not be what the authors have wished to convey, but this is the message that came accross to me when I read it. On the contrary, the litterature on vaccine hesitancy has underlined the many reasonable reasons people can have to distrust authorities (ranging from discriminations, past scandals, politicization of vaccines…). I suggest the authors read Heidi Larson’s review of the literature on the role of trust (in Human Vaccines and Immunotherapeutics). Not all form of distrust of authorities constitutes conspiracy beliefs. The term trust does not appear even once in this paper. This should be especially clear in the context of the COVID pandemic : some governments’ actions have given good reasons to be cautious : Trump politicized the issue, the UK government decided to space the second dose against scientific advice, the contracts signed between the EU and pharmaceutical companies have not been made public… This focus on conspiracy beliefs can come from the fact that the questionnaire did not offer any items where respondents could express forms of distrust of various actors involved in vaccine production and rollout in Jordan (apart for the conspiracy beliefs discussed previously). This should be clearly adressed.
Response: We would like to thank the reviewer for giving us the opportunity to clarify this issue. In the VCBS score used in this study, the items assessed -at least partially- the level of trust in pharmaceutical companies and governments using the following items: Pharmaceutical companies cover up the dangers of COVID-19 vaccines; The government is trying to cover up the link between vaccines and autism. (Please refer to Supplementary File S1)
However, we added the following paragraph to the Discussion section (Page 11, Lines 371-374) based on the reviewer comment,: “An additional important factor that should not be overlooked in the context of COVID-19 vaccine hesitancy is the role of mistrust in policymakers, healthcare professionals and vaccine providers [65]. These factors should be considered in further studies addressing vaccine hesitancy in the country.”
- More information should be provided on the history/context of vaccine hesitancy in Jordan before COVID-19 but also on the debates over COVID-19 vaccination. Have their been major public debates on the safety of these vaccines in the Jordan press ? Are there important antivaccine movements in Jordan and have they made themselves heard since the beginning of the pandemic ? This is important to interpret the results. In the discussion the authors write this : « First, University students in Jordan showed an overall low intent to get COVID-19 287 vaccines (34.9%). This alarming result is in agreement with our recent survey that inves-288 tigated COVID-19 vaccine hesitancy among the general public in Arab countries, where 289 the vaccine acceptance rate was merely 28.4% in Jordan [23]. » This should be developped so we can get a grasp of the specificities of attitudes towards vaccines in Jordan. Indeed, vaccine hesitancy is very context specific, and providing more context on Jordan would be -in itself – a welcome addition to the littérature.
Response: We would like to thank the reviewer for this important comment. In Jordan, there is no organized anti-vaccination movement to the best of our knowledge. However, circulating information on different media outlets (TV, newspapers, social media platforms), pointed to the presence of individuals who advocate rejection of COVID-19 vaccines citing unreliable sources to show that these vaccines are unsafe and can cause death (https://www.alaraby.co.uk/society/%D8%A7%D9%84%D8%A3%D8%B1%D8%AF%D9%86%D9%8A%D9%88%D9%86-%D9%8A%D9%86%D8%A8%D8%B0%D9%88%D9%86-%D9%84%D9%82%D8%A7%D8%AD-%D9%83%D9%88%D8%B1%D9%88%D9%86%D8%A7).
However, the scope of vaccine hesitancy in the country has not been investigated thoroughly in literature which is manifested by limited number of publications on this topic, and was highlighted by Larson et al, which showed under representation of low- and middle-income countries in the review: “Measuring trust in vaccination: A systematic review, Hum Vaccin Immunother. 2018”.
Additionally, the issue of COVID-19 vaccine acceptance has been a subject of debate in the local media outlets, but without scientific evidence. Examples of such articles can be seen in the following news articles from the country.
https://alghad.com/?p=969725
https://alghad.com/?p=967996
Thus, we believe that our results can be helpful to explore the scope of COVID-19 vaccine hesitancy in the country. For the elaboration of the general attitude towards vaccination in the country, the scarcity of research investigating this topic precluded further discussion as requested by the reviewer.
- Here are some more minor comments :
The definition of vaccine hesitancy provided in the introduction (« The refusal to accept vaccination in spite of effective and safe vaccine availability 71 defines vaccine hesitancy ») does not correspond to the standard in the littérature. Hesitant people often do get vaccinated and hesitancy also arises in contexts where there is uncertainty regarding the effectiveness and safety of a given vaccine (as is the case regarding COVID-19 vaccination). I suggest the authors turn to the standard definitions provided by larson, dubé or Peretti-Watel.
Response: We would like to thank the reviewer for this important comment, and accordingly, we modified the definition based on “Peretti-Watel, P., Larson, H. J., Ward, J. K., Schulz, W. S., & Verger, P. (2015). Vaccine hesitancy: clarifying a theoretical framework for an ambiguous notion. PLoS currents, 7, ecurrents.outbreaks.6844c80ff9f5b273f34c91f71b7fc289. https://doi.org/10.1371/currents.outbreaks.6844c80ff9f5b273f34c91f71b7fc289”
The new paragraph in the Introduction section (Page 2, Lines 71-74): “The “delay in acceptance or refusal of vaccines despite availability of vaccination services” defines vaccine hesitancy, which is a complex and context-specific phenomenon that varies across time, place and vaccines [27-29].”
- I would strengthen the justification for focusing on students. In the paper, they are presented as « insightful », etc. I don’t think this really underlines the importance of studying this sub-group. In the discussion, the authors mention their potential influence on fellow young people : this could be expanded (especially since this is more likely to be the case for medical students than for students of the humanities, but I’m not familiar with social status in Jordan). Other justifications could be that young people tend to be more reluctant to vaccinate against covid and that it’s important to understand why because eventually we will need to get them vaccinated to avoid the spread of the virus.
Response: We would like to thank the reviewer for this important comment. Based on this comment, we added the following paragraph to the Discussion section (Page 9, Lines 285-288): “In addition, previous evidence showed that University students can comprise a core group that would be helpful in addressing vaccine hesitancy through promoting a positive attitude towards vaccination [51].”
- I think a little bit more should be said about the potential biases due to the chain-referral sampling
Response: We would like to thank the reviewer for this important comment, and based on this comment, we added the following paragraph to the limitations sub-section (Page 11, Lines 356-369): “Despite the advantages of chain-referral approach in sampling of hidden populations, an obvious caveat of this approach is that bias introduced by the first accessed participants would be inevitable. Thus, generalizability of the results cannot be ensured.”
Reviewer 2 Report
This is a timely and important study to help understand vaccine hesitancy among young adults during the current pandemic. I did not see reporting of IRB approval for this study, so I have ethical concerns. I also feel the authors' writing style is at times strongly stated or "emotional" and needs to be neutral.
Language throughout at times is not scholarly. For example the title of the manuscript states "never-ending story" I would suggest removing the last part of the title it biases the reader.
In the discussion section section could the authors make any connection to the sample's age group which has not encountered many of the the infectious diseases of the past that older generations have but are no longer prevalent due to vaccination resulting in less fear of infectious diseases.
In the discussion section I would have like to seen research implications from the findings of this study. Do you feel a qualitative study design may help to understand the beliefs of this population more accurately? Young females were the largest group to not want vaccination....could it be due to fear of infertility. Also, there was a high percentage of respondents who did not answer yes or no to survey questions and this warrants deeper exploration.
Author Response
Reviewer #2 comments
- This is a timely and important study to help understand vaccine hesitancy among young adults during the current pandemic. I did not see reporting of IRB approval for this study, so I have ethical concerns.
Response: I would like to thank the reviewer for the important comment. As mentioned in the declarations section: “The study was approved by the Department of Pathology, Microbiology and Forensic Medicine (meeting 03/2020/2021 that was held on 07-01-2021), and by the Scientific Research Committee at the School of Medicine/University of Jordan (reference number: 321/2021/67).”
We also added the following sub-section into the Materials and Methods section (Page 4, Lines 164-170):
2.4. Ethical considerations
The study was approved by the Department of Pathology, Microbiology and Forensic Medicine (meeting 03/2020/2021), and by the Scientific Research Committee at the School of Medicine/University of Jordan (reference number: 321/2021/67). An informed consent was ensured by the presence of an introductory section of survey used in this study, with a mandatory question asking for agreement from the respondent to participate in the survey study. All collected data were treated with confidentiality.
- I also feel the authors' writing style is at times strongly stated or "emotional" and needs to be neutral.
Language throughout at times is not scholarly. For example the title of the manuscript states "never-ending story" I would suggest removing the last part of the title it biases the reader.
Response: We would like to thank the reviewer for this comment, and based on this comment and the comment from reviewer #3, we modified the title as follows: “Low COVID-19 Vaccine Acceptance is Correlated with Conspiracy Beliefs among University Students in Jordan”
- In the discussion section section could the authors make any connection to the sample's age group which has not encountered many of the the infectious diseases of the past that older generations have but are no longer prevalent due to vaccination resulting in less fear of infectious diseases.
Response: We would like to thank the reviewer for this important comment, and based on this comment and those of the first reviewer, we added this comment to the Discussion section (Page 10, Lines 311-316): “Low perception of disease risk can be an important determinant in the declining intention to get the vaccines, as shown recently by Caserotti et al [53]. Unravelling such an attitude would be highly valuable to tailor communication regarding COVID-19 vaccination [54]. This would be particularly relevant in countries with a prominent youth bulge (e.g. Jordan, where 20% of its population aged 20-29 in 2019) [55].”
- In the discussion section I would have like to seen research implications from the findings of this study. Do you feel a qualitative study design may help to understand the beliefs of this population more accurately? Young females were the largest group to not want vaccination....could it be due to fear of infertility. Also, there was a high percentage of respondents who did not answer yes or no to survey questions and this warrants deeper exploration.
Response: We believe that the results of the current study can be of value in helping public health efforts in the country and the region to tackle the issue of COVID-19 vaccine hesitancy, and might advocate the importance of increasing the awareness of University students regarding vaccination in general and the role that COVID-19 vaccines can play to control the ongoing pandemic. Regarding the fear of infertility in relation to sex, it was not mentioned in the results since it did not yield a statistically significant difference (p = 0.125). Please refer to the Table below
|
|
Sex |
||||
|
Male |
Female |
||||
|
Count |
Column N % |
Count |
Column N % |
||
|
COVID-19 vaccination will lead to infertility |
Yes |
15 |
4.9% |
33 |
4.1% |
|
No |
179 |
58.9% |
425 |
53.0% |
|
|
Maybe |
110 |
36.2% |
344 |
42.9% |
|
For the respondents who did not answer yes or no (maybe/no opinion), the rationale behind stratification of such respondents with the “COVID-19 vaccine hesitancy” group vs. the vaccine acceptance group was mentioned in the response to reviewer #1 (please refer to our response to point #2)
Reviewer 3 Report
The authors well conducted the research study on conspiracy beliefs and COVID-19 vaccine acceptance. In the title, the part "Never-ending Story of COVID-19 and Conspiracy" seems referring to results that are not in the study actually. How can you state that in the title? May be something more informative could be appropriate, i.e. "Low COVID-19 Vaccine Acceptance is Correlated with Conspiracy Beliefs among University Students in Jordan, January 2021".
In lines 290-293, the authors outlined the relatively higher vaccine acceptance rate among students in Health Schools (43.5%), compared to their peers in other topics. This can be related not only to their higher knowledge about the disease, but also to their specific ability to understand and trust the results on safety and efficacy of the clinical trials of the COVID-19 vaccines.
The participation to the study by using an online-based survey distributed by social media platforms biased the study results towards those students who can have easy access to social media and trust them. You should discuss this limit in lines 334-335, also considering: what is the proportion of the students using the internet in Jordan, and which one of those using social media? Can you identify how many participants responded through Facebook, rather than Twetter or WhatsApp, etc.?
In table 1, abround 10% of the students have "History of chronic disease". Does this proportion well reflect the real percentage of chronic patients among students in Jordan? you can use this variable as an indicator of rapresentativeness of the survey results. You should also comment on the fact that the participants did not reach the minimum calculated sample size (1064 respondents).
For clarity, please add the totals to the tables (line and row).
Author Response
Reviewer #3 comment
- The authors well conducted the research study on conspiracy beliefs and COVID-19 vaccine acceptance. In the title, the part "Never-ending Story of COVID-19 and Conspiracy" seems referring to results that are not in the study actually. How can you state that in the title? May be something more informative could be appropriate, i.e. "Low COVID-19 Vaccine Acceptance is Correlated with Conspiracy Beliefs among University Students in Jordan, January 2021".
Response: We would like to thank the reviewer for this comment, and based on this comment and the comment from reviewer #2, we modified the title as follows: “Low COVID-19 Vaccine Acceptance is Correlated with Conspiracy Beliefs among University Students in Jordan”
- In lines 290-293, the authors outlined the relatively higher vaccine acceptance rate among students in Health Schools (43.5%), compared to their peers in other topics. This can be related not only to their higher knowledge about the disease, but also to their specific ability to understand and trust the results on safety and efficacy of the clinical trials of the COVID-19 vaccines.
Response: We would like to thank the reviewer for pointing to this important aspect that we did not discussed before. Thus, we modified the discussion based on the reviewer comment as follows Discussion section (Page 10, Lines 306-308): “Additionally, students in Health Schools might have a better ability to fathom the results of clinical trials on COVID-19 vaccines; thus, having a higher trust and acceptance of such novel vaccines.”
- The participation to the study by using an online-based survey distributed by social media platforms biased the study results towards those students who can have easy access to social media and trust them. You should discuss this limit in lines 334-335, also considering: what is the proportion of the students using the internet in Jordan, and which one of those using social media? Can you identify how many participants responded through Facebook, rather than Twetter or WhatsApp, etc.?
Response: We would like to thank the reviewer for this relevant comment.
The proportions of using the internet appeared high in the studies and reports issued in the last five years (Net-Med Youth Project: Jordan Youth Media Perception Survey, Ages 18 -29, Administered among Universities, Community Colleges and Households. Commissioned under the Networks of Mediterranean Youth Project (NET-MED Youth), which is funded by the European Union and implemented by UNESCO; link: http://www.unesco.org/new/fileadmin/MULTIMEDIA/FIELD/Amman/pdf/Jordan_Youth_Media_Perception_SurveyEN.pdf); Almarabeh, T. , Majdalawi, Y. and Mohammad, H. (2016) Internet Usage, Challenges, and Attitudes among University Students: Case Study of the University of Jordan. Journal of Software Engineering and Applications, 9, 577-587. doi: 10.4236/jsea.2016.912039.
However, it is not possible to track back the proportion of participants who responded through each social media platform invitation.
Based on the reviewer’s comment, we added the following statement to the limitations sub-section (Page 11, Lines 353-354): “Limitations of this study involve potential sampling bias that could be related to coverage of the students who use the internet and social media platforms regularly”.
- In table 1, abround 10% of the students have "History of chronic disease". Does this proportion well reflect the real percentage of chronic patients among students in Jordan? you can use this variable as an indicator of rapresentativeness of the survey results. You should also comment on the fact that the participants did not reach the minimum calculated sample size (1064 respondents).
Response: We would like to thank the reviewer for mentioning this precise and valuable point. In the item of the questionnaire that was used to assess the history of chronic disease, the exact wording was as follows: “Do you suffer from any chronic diseases (such as diabetes, allergy, hypertension or heart disease)?”. We added this complete item to the Materials and Methods section (Page 3, Line 124)
Thus, allergy was one of these conditions included, and as an example, a previous study from Jordan, showed that asthma was moderately common in Jordan (Abu-Ekteish, F.; Otoom, S.; Shehabi, I. Prevalence of asthma in Jordan: comparison between Bedouins and urban schoolchildren using the International Study of Asthma and Allergies in Childhood phase III protocol. Allergy Asthma Proc 2009, 30, 181-185, doi:10.2500/aap.2009.30.3208). However, the previous study was conducted among school children, even though it can give a hint about the prevalence of some allergic disease in the country. Lack of robust and recent data on the prevalence of other chronic disease among university students in Jordan precluded further assessment of the representativeness of the study sample, even though we mentioned the potential sampling bias as a limitation of this study.
For the minimum calculated sample size, I believe that we reached the minimum as the total number of participants included in final analysis was 1106 students as mentioned in the results section (Page 4, Line 181).
- For clarity, please add the totals to the tables (line and row).
Response: We have to clarify one issue that has already been present in the Materials and Methods section (Page 3, Lines 136-137): “Response to all items before submission was mandatory to eliminate the effects of item non-response.” Hence, we believe that adding the totals to the tables will be of minimal value and we prefer to keep the tables in the current format.
We are deeply grateful for the comprehensive, insightful and thorough review.
Round 2
Reviewer 1 Report
OK